



# Automating the Analysis of Hailstone Layers

Joshua S. Soderholm[1] and Matthew R. Kumjian[2]

[1]Science and Innovation Group, Australian Bureau of Meteorology, Docklands, Victoria, Australia
[2]Department of Meteorology and Atmospheric Science, The Pennsylvania State University, University Park, Pennsylvania

**Correspondence:** Joshua S Soderholm (joshua.soderholm@bom.gov.au)

**Abstract.**

The layered structures inside hailstones provide a direct indication of their shape and properties at various stages during growth. Given the myriad of different trajectories that can exist, and the sensitivity of rime deposit type to environmental conditions, it must be expected that many different perturbations of hailstone properties occur within a single hailstorm; however, some commonalities are likely in the shared early stages of growth, for hailstones of similar size (especially those that grow along similar trajectories) and final growth near the melting level. It remains challenging to extract this information from a large sample of hailstones because of the time required to prepare cross sections and accurately measure individual layers. To reduce the labour and potential errors introduced by manual analysis of hailstones, an automated method for measuring layers from cross section photographs is introduced and applied to a set of hailstones collected in Melbourne, Australia. This work is motivated by new hail growth simulation tools that model the growth of layers within individual hailstones, for which accurate measurements of observed hailstone cross sections can be applied as validation. A first look at this new type of evaluation for hail growth simulations is demonstrated.

## 1 Introduction

The internal structures and composition of natural hailstones can provide remarkable insights into their growth evolution and the associated in-storm conditions, akin to climate reconstructions from paleo-proxies. One of the most striking features of cross sections extracted from hailstones is the layering of clear, glaze ice with milky, rime ice. These two types of ice deposits were first described by Clark (1946) from Mount Washington Observatory (US) experiments of rime icing, and later adapted for hailstones by Weickmann (1953), Macklin (1962), and Browning (1966). Rime ice opacity is dependent on the size distribution of trapped air bubbles, whereby highly concentrated minute air bubbles lead to multiple scattering of light, producing more opaque ice, in contrast to clear glaze ice, which is largely free of small air bubbles (Browning and Beimers, 1967). Small air bubbles form when collected supercooled droplets freeze near instantaneously on the hailstone surface leaving the hailstone surface dry, and thereby not permitting sufficient time for dissolved or trapped air to escape. Further, Carras and Macklin (1975) showed that the concentration of air bubbles was a function of the freezing rate, thus providing a quantitative indication of the dry growth conditions. The freezing of collected supercooled droplets also releases latent heat to the hailstone owing to the enthalpy of freezing. If this excess thermal energy cannot be transferred to the ambient environment while maintaining a surface temperature of less than 0 °C, the hailstone surface will become wet. Under this wet growth regime, trapped and





dissolved air has time to escape during freezing, leaving mostly transparent glaze ice. Additionally, wet hailstones can also grow as a mixture of solid ice and excess liquid water, known as spongy growth, evident in completely frozen hailstones as transparent ice with fine, hair-like chains of elongated bubbles (Knight and Knight, 1968).

The separation between dry and wet growth regimes is often marked by abrupt changes in ice opacity, rather than a more gradual transition. This is the result of the freezing rate, which is highly sensitive to hailstone surface temperatures near 0 °C (Carras and Macklin, 1975). Carte and Kidder (1966) used these abrupt transitions to manually identify layers within 673 hailstones collected across 43 days in South Africa, providing information about the number of layers and the dry growth contribution. A similar analysis was performed in other studies, including Browning and Beimers (1967) and Knight and Knight

(1970). Compared with other forms of hailstone analysis (e.g., water isotopes, crystal structure and air bubble concentration), hailstone layers only provide a broad indication of conditions within the growth environment[1]. However, the contribution of wet and dry growth regimes, evident as layers, can be simulated by hail growth models, and therefore observed hailstone layers can be used to evaluate this simulation output. This approach was first demonstrated by Ziegler et al. (1983), whereby the physical structure of a small set of observed and modelled hailstones was shown to be strikingly similar. Recent advances in

hail growth modelling (e.g., Kumjian and Lombardo, 2020; Adams-Selin, 2022) and trajectory simulations by Brook et al. (2021), coupled with high-performance computing permits simulation of millions of individual hailstones within a hailstorm. Validation of modelled growth processes is difficult as typically only hail size reports are available for simulated cases. Evaluation of modelled hailstones properties with large samples of observed hail will provide new insights into the ability of these new simulation tools to accurately model the complex growth processes, including transitions between dry and wet growth

regimes.

      Motivated by the need to validate modelled hail growth processes using observations, this paper demonstrates a novel technique for automating the measurement of opaque layers produced by dry growth from hailstone cross section images. The primary aim of this technique is to make processing large samples of hailstones more feasible by providing objective and reproducible measurements, and by reducing the labour required to manually analyse cross sections. Drawing from recent

advances in the field of dendrochronology to automate the analysis of tree rings (e.g., Cerda et al., 2007), computer vision techniques are applied to extract the 2D geometry of individual growth layers in hailstone cross sections. This paper details the sample preparation, imagery capture, automated layer analysis technique, and outputs. Results for a collection of hailstones from a hailstorm event in Melbourne, Australia on 19 January 2020 are discussed and expected applications are identified. A comparison of bulk statistics between the Melbourne hail collection and modelled hailstones from an idealised simulation are

also shown to demonstrate the insights gained for evaluating simulation tools.

## 2    Data and Approach

Cross sections presented within this paper were extracted from hailstones collected during a hailstorm event through the eastern suburbs of Melbourne on 19 January 2020. Forty unbroken hailstones were collected within a 3m x 3m area during the event

---

[1]these retrievals have questionable value given the large number of assumptions required (List, 1977)





(37.84° S 145.06° E) and immediately bagged and stored in a freezer at a temperature of -18 °C. The maximum dimension of hailstones ranged from 23.5 mm to 60 mm, and the appearance varied from milky larger stones with many small lobes to oblate, partly melted stones (Fig. 1). The variety of shapes and sizes are indicative of multiple growth trajectories within the storm. The Melbourne hailstorm was part of a larger outbreak across Eastern Australia between the 19-21 January 2020, which incurred an insurance industry loss of more than $1.8B AUD (PERILS, 2021).

## 2.1 Sample Preparation and Imaging

Nondestructive measurements were first performed before cuts were made to extract a cross section. This included dimension measurements using calipers, hailstone mass, and photogrammetry scanning of larger stones to compile a digital 3D model. To extract and photograph a cross section, hailstones were warmed to just below melting point and transferred to a cool room (approximately 4 °C). A hot-wire tool and cutting guide was used to perform slices. The first cut was made approximately intersecting the centre of the stone and orientated normal to the minor axis. The two hemispheres were then inspected to determine where the embryo was present. If the embryo centre was located more than approximately 3 mm below the surface, the cut face was melted on an unheated metal plate to remove excess ice. A second cut was made through the hemisphere containing the larger portion of the embryo, producing a cross section on the order of 2-3 mm thick.

Once cut, a cross section was immediately mounted onto a large glass slide and placed inside a light tent (which provides uniform illumination of the sample) with a black background to enhance the contrast between transparent and opaque layers. Some minor melting occurs at the cool room temperature (4 °C; however, the liquid water coating on the cross section was found to fill any surface defects and therefore be beneficial for the photography. Condensation on the glass slide was avoided by using an alcohol based anti-fog coating. An 18 megapixel DSLR camera with an 18-55 mm lens was mounted above the light tent to capture photos. Additional photos that included a measurement ruler were also taken with the cross sections to provide a reference for the pixel size. For each cross section photo, the hailstone embryo centre (if present), embryo outline (if present), and reference measurement were annotated using the VGG Image Annotator (VIA) tool (Dutta and Zisserman (2019); Fig. 2a). Performing this initial step manually was important as the automated analysis requires an accurate embryo centre and pixel size.

## 2.2 Layer Analysis Technique

Cross section photos were prepared for analysis by replacing the background with the colour black (using the raster editing software package GIMP; The GIMP Development Team 2019). Prepared images were then converted into the Hue-Saturation-Lightness (HSL) color space to utilise the lightness information for separating layers (range of 0-255). Similarly, the HSL color space was used by (Soderholm et al., 2020) to isolate individual hailstones in aerial imagery for a computer vision assessment of the hail size distribution. The lightness field of each cross section image was then stretched to fill the entire range. This stretched lightness field maximises the contrast across the colourless hailstone layers, increasing the separation between layers. Using the annotated reference measurement, each image was resized such that 30 pixels represent 1 mm. Finally, a Gaussian filter was applied to minimise the appearance of cracks and small features (e.g., radial bubbles) not associated with growth





layers. A filter standard deviation of 4 pixels was manually determined to be most effective for minimising these artefacts without excessive smoothing of layer boundaries. For the purpose of this description, a 'layer' was defined as an opaque ice layer associated with the dry growth regime. A conceptual diagram of the methods is provided in Fig. 2 and direct outputs from the automated detection are shown in Fig. 3.

The first step in the analysis involves the construction of 72 evenly spaced radial transects (5° interval) spoked from the embryo centre, anti-clockwise from the positive x-axis. Pixel values (lightness intensity) were extracted at a constant distance interval (1 pixel length) along each transect. This use of polar coordinates exploits the approximately circular symmetry of hailstone layers (Browning, 1966). For each radial transect, local lightness maxima were identified using the SciPy find_peaks function (Virtanen et al., 2020), which performs a simple comparison of neighboring values. The find_peaks function parameters were set such that local maxima must exceed 80 (lightness), have a prominence of 25 (lightness) from adjacent local minima and be separated by at least 2 mm from other maxima. Optimisation of these filter parameters was performed manually across the entire set of cross sections. Fig. 2b demonstrates local peaks associated with layers in a single radial transect. The width of each layer was then identified where the lightness value falls below 30% of the local maximum value either side of the peak (Fig. 2c and 3a). Overlapping layers were then merged and thin layers near the edge (less than 1 mm from the edge and layer) were removed to avoid reflection artefacts produced by the water film along the edges of hailstones. A visualisation of candidate layers across all transects using the azimuth-range space is shown in Fig. 3b.

The second step in the analysis involves the consolidation of candidate layers identified from the 72 radial transects into a single set of layers for the hailstone. Contiguous features in the azimuth-range space with an azimuthal width of less than 30° are assumed to be not associated with layers, such as large bubbles, and are therefore removed (shown as yellow regions in Fig. 3b). Next, the 72 radial transects were consolidated into 1 radial transect by counting the number of times each range bin was assigned as a layer across all azimuths (Fig. 2d, Fig. 3c). This approach allows individual layers to be separated and the area of each layer to be calculated. To achieve this, the consolidated radial transect was first smoothed using a 10-pixel moving average filter to reduce noise from spurious opaque features. The SciPy find_peaks function was then applied to identify layers by separating local maxima along the smoothed transect. Parameters of the find_peaks were manually optimised to capture both fine and wide layers such that the local maxima must have a prominence of 10 (bin count) from adjacent local minima and be separated by at least 2 mm from other maxima. Finally, the area and area-weighted radius of each layer were calculated.

Using the area and area-weighted layer radius, it is possible to construct the equivalent circular cross section of the hailstone that preserves the layer area and radial distance, regardless if its symmetry (Fig. 2e, Fig. 3d). This output provides a direct comparison for explicit simulations of hailstone growth (e.g., Ziegler et al. 1983 Fig. 24, 25; Kumjian and Lombardo 2020 Fig. 3).

## 3 Application

The hailstone collection from the 19 January 2020 Melbourne hailstorm provides an opportunity to demonstrate the practical application of the layer analysis technique (LAT). A cross section sample was prepared for each of the forty hailstones and an-





notated photographs were compiled according to the procedure described in section 2. A composite image of all cross sections is shown in Fig. 4 using the uniform pixel size of 1/30 mm. Inspection of the hailstone cross sections reveals a remarkable diversity of structures, including conical graupel embryos (e.g, hailstones 3, 14, 21), possible frozen droplet embryos (e.g., hailstones 2, 5, 25), radial bubbles (e.g., hailstones 8, 12, 16) and hyperfine growth layers (e.g., edges of hailstones 1, 18, 39). The LAT was applied to each image, and a composite of the respective equivalent circular cross sections is shown in Fig. 5

using a same scaling as in Fig. 4. Overall, the LAT performs well across the sample of cross sections. Some minor issues are apparent where semi-opaque ice is present, especially in the outer regions of the hailstones (e.g., hailstones 5, 22). Further, the LAT will often merge layers which were overlapping (hailstones 3, 15) or were very thin with diffuse edges (hailstones 6, 21); however, given the conservation of the layer area and radius in the consolidation procedure, the impact on the equivalent circular cross section is minimal.

Statistics generated from the LAT for total wet growth fraction and the final wet growth layer fraction are combined with manual measurements of maximum dimension and minor-to-major axis ratio, and derived equivalent dimension, to investigate hailstone properties (Fig. 6). Equivalent dimension is calculated from an oblate spheroid model using the measured intermediate and major dimensions to provide direct comparison with the diameter of simulated hailstones. Ideally an oblate ellipsoid model should be used, but the minimum dimension was not measured for 14 of the 40 hailstones. Investigation of wet growth metrics

is motivated by observations of significant wet growth, especially as an outer layer, in large hailstones (Knight and Knight, 2005; Kumjian et al., 2020). A slight shift towards more nonspherical hailstones with increasing maximum dimension size is apparent in Fig. 6a. The same decreasing trend and similar axis ratios have been shown in studies with larger samples of hailstones (e.g., Knight, 1986; Shedd et al., 2021). The total wet growth fraction remains remarkably consistent across the sample of hailstones with varying axis ratios and size (Fig. 6b,c), with 68% of samples having between 50% and 70% wet

growth. A significant portion of this wet growth occurs in the final wet layer, which contributes to more than 30% of the total wet growth area for more than 71% of samples. Further, the final wet growth layer fractions were largest for smaller hail sizes (Fig. 6d), indicating that regardless of size and shape, the final stage of wet growth as hailstones approached the melting level was a significant contribution. Some caution must be placed on interpreting these findings due to the small sample size in this study and the impact of melting on the outer layers during descent.

To provide a first look at evaluating a hail growth model using the outputs from the LAT, the 'umax31' storm from Kumjian and Lombardo (2020) was used to simulate growth layers in a sample of hailstones for comparison with the observed data. Note that this comparison is simply of the growth layer bulk statistics; this simulation is from a highly idealised case and is not representative in any way of the Melbourne event. The sample of simulated hailstones was selected to approximately match the number and sizes of observed hailstones from the Melbourne case. To achieve this, the number of observed hailstones in 5 mm

equivalent dimension intervals (20-25 mm, 25-30 mm, etc.) was first determined. Then, all simulated hailstones from Kumjian and Lombardo (2020) (using a 5-mm embryo) within a specified interval were identified and a random number generator (without repeats) was used to select from each size class a number of hailstones matching the number in the observed size class (Fig. 7). If there were no simulated hailstones within a given size interval, the next embryo size up was used (7.5 mm, then 10.0 mm). For the larger size classes, too few large hailstones were simulated, so any stone with a size greater than 50 mm was



selected randomly from the simulated population. In contrast to the observed Melbourne hailstones, wet growth dominated the collection of hailstones from the simulated 'umax31' storm (Fig. 8). The mean wet growth fraction was 88% for the simulated hailstones, much higher than the observed samples (63%), with many simulated hailstones growing only in the wet regime (Fig. 9).

This apparent excessive wet growth in the simulations highlights possible limitations of the modelling approach. Entering

the wet growth regime requires large collection rates (a factor of hailstone size, fall speed, and cloud liquid water content) and the inability to dissipate excess thermal energy to the environment. Each of these may contribute to the discrepancies between the simulated and observed hailstone properties. For example, large uncertainties exist in hailstone size-fallspeed relationships (e.g., Heymsfield et al., 2018, 2020); positively biased fallspeeds for hailstones of a given size would lead to positively biased collection rates. However, such high-biased fallspeeds could reduce residence time in the hail growth region, possibly limiting

growth. Additionally, simulated cloud liquid water contents may be too large, especially given the rather moist sounding used in the 'umax31' simulation, resulting in larger collection rates than observed in the Melbourne case. Further, the thermal energy transfer (which is parameterized based on Rasmussen and Heymsfield (1987) for "rough" spherical hailstones) may be too inefficient. The observed hailstones exhibit more complex geometries, which could enhance thermal energy transfer by (i) an increased surface area from which thermal energy may be conducted away, and (ii) creating greater turbulence in the

hailstone's wake, which efficiently transfers thermal energy away from the hailstone. To further understand the impact of these factors, a simulation of the Melbourne hailstorm and hailstones would be required for direct evaluation using the observed hail collection, and more sophisticated treatment of the growth processes for realistic hailstones is necessary.

## 4  Summary and Outlook

Computer vision provides a powerful tool for automating the analysis of hailstone cross sections. Automation not only min-

imises the possibility of human error and the time required for manual measurements, it also permits measurement of individual layer area and thereby the reconstruction of equivalent circular cross sections for comparison against simulation outputs. Application of the LAT to a small collection of 40 hailstones from the 19 January 2020 Melbourne hailstorm event demonstrates the technique robustly captures layers. Statistics generated from the LAT show that despite the varying sizes and shapes, the total wet growth fraction was remarkably consistent across a majority of samples. Further, a significant portion of this wet growth

occurred in the final layer, especially for smaller hailstones, highlighting the importance of this final growth unit. Comparison of bulk statistics from the Melbourne collection against hail modelled in the idealised 'umax31' storm from Kumjian and Lombardo (2020) provided a first look at how the LAT outputs can be used to evaluate simulations, and highlighted a potential bias towards wet growth in the simulation.

Care must be taken when applying the LAT to new hailstone collections; the cross section preparation, photography and

image processing steps are critical to ensure consistent results. Further, changes to the image pixel size or lightness range would require careful review of manually optimised parameters. Considering these factors, the LAT could also be applied to digitise existing collections of hailstone cross sections. Future work to investigate the evolution of hailstone shape during



growth using additional information on layer geometry extracted by the LAT is planned. This information, coupled with the embryo type and size, is expected to provide further insights into hailstone growth. The transmitted light photography used to

capture cross sections in Fig. 4 is often complemented with photographs that use cross-polarised light for examining the ice crystals, which can be used to infer growth conditions and further constrain changes in growth regime (Macklin et al., 1977; Pflaum, 1984). Application of computer vision for the automation of layer measurements from ice crystal changes will be explored further. Hailstone structure observations are anticipated to become increasingly important with the development of new simulation and radar-based approaches for modelling hailstone growth and trajectories. To achieve this goal, we advocate

for much larger collections of hailstones (ideally, hundreds of hailstones) within the coverage of observational networks and representative simulations of the parent hailstorm and individual hailstones be performed.

*Code availability.* Code to generate the LAT analysis and figures in the paper have been provided via the v1.0.0 tagged release of the hail-xsection repository https://github.com/joshua-wx/hail_xsection/releases/tag/v1.0.0

*Data availability.* The cross section photos and hail growth and trajectory model outputs used to produce figures and analysis presented in

this article are available at https://doi.org/10.5281/zenodo.6831306

*Author contributions.* JSS performed the hail cross section preparation, designed the methodology and developed the analysis code. MRK supported the statistics analysis and performed the simulations of hail growth. JSS prepared the manuscript with contributions from MRK.

*Competing interests.* The authors declare that they have no conflict of interest.

*Acknowledgements.* The authors wish to thank the assistance of Chen Li for collecting and preserving hailstones during the Melbourne

hailstorm and Julian Brimelow for input during the drafting of the manuscript. MRK was supported by a grant from the U.S. National Science Foundation (AGS-1855063) and an award from the Insurance Institute for Business and Home Safety.





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



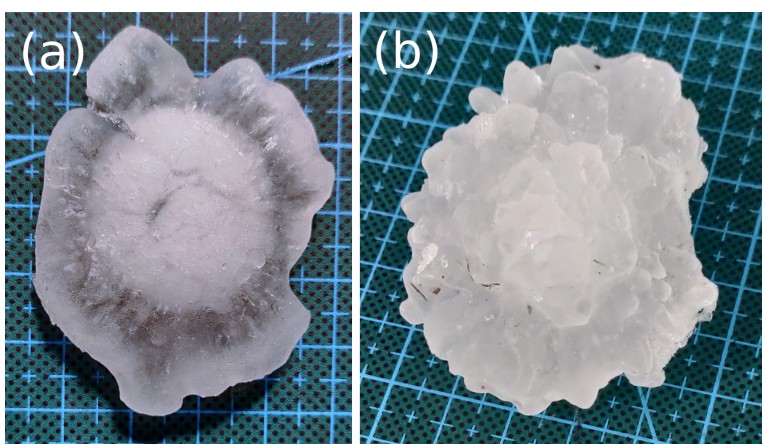

**Figure 1.** Examples of the two main hailstone shapes observed: (a) highly oblate, partly melted hailstones without lobes, (b) larger, approximately-spherical opaque hailstones covered in many small lobes. Photographs were captured prior to the extraction of cross sections. These hailstones are identified as numbers 5 and 4 respectively in subsequent figures.



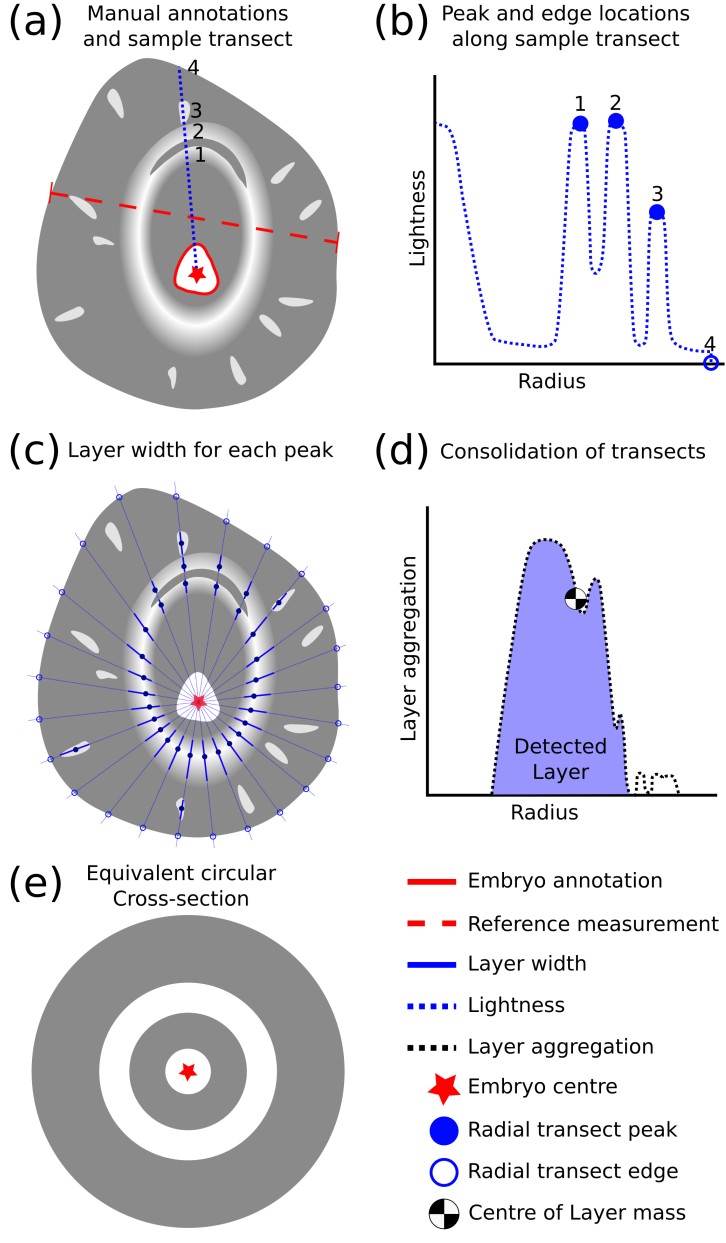

**Figure 2.** Conceptual diagram of analysis procedure: (a) manual annotations and sample transect with 3 peaks marked (1,2,3) and the edge location (4), (b) Representation of transect in radius-lightness space with peaks and edge marked from (a), (c) Measurement of layer width for each peak detected, (d) detected layer centre of mass from consolidated transect analysis, (e) Equivalent circular hailstone cross section where wet and dry growth is represented by gray and white shading, respectively.




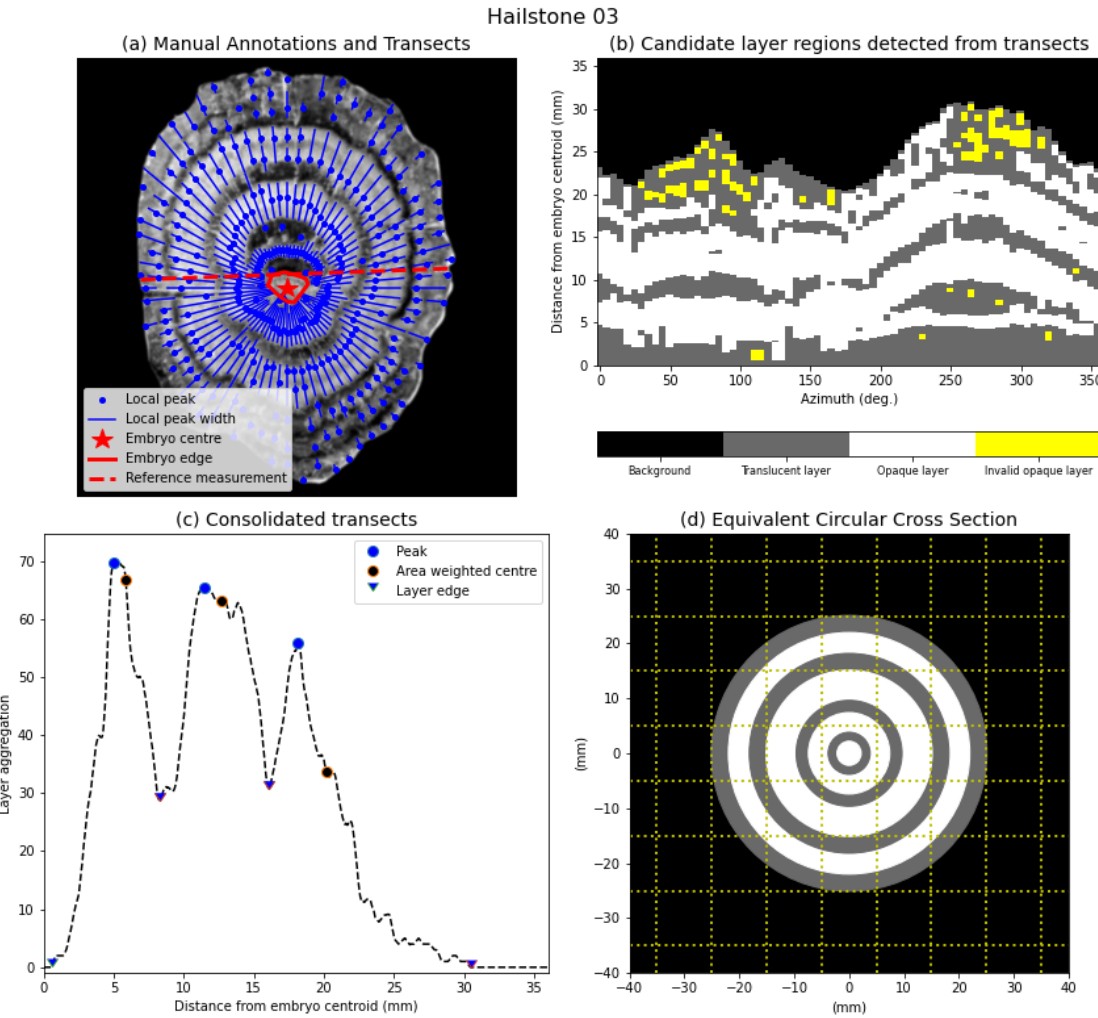

**Figure 3.** Demonstration of analysis procedure for hailstone 3: (a) manual annotations (red elements) and layer peak detection along radial transects (blue elements), (b) Transect analysis shown in the azimuth-range space, (c) Detection of layers from consolidated transect analysis, (d) Equivalent circular hailstone cross section where wet and dry growth is represented by gray and white shading respectively.







**Figure 4.** Hailstone cross sections from the Melbourne hailstorm event (19 January 2020). Grid lines shown at a 1 cm spacing.





**Figure 5.** Equivalent circular hailstone cross sections generated from the layer analysis technique. White-shaded regions represents opaque dry-growth layers and gray shaded regions indicate translucent wet-growth layers. Grid lines shown at a 1 cm spacing starting from 0.5 cm from the embryo centroid.




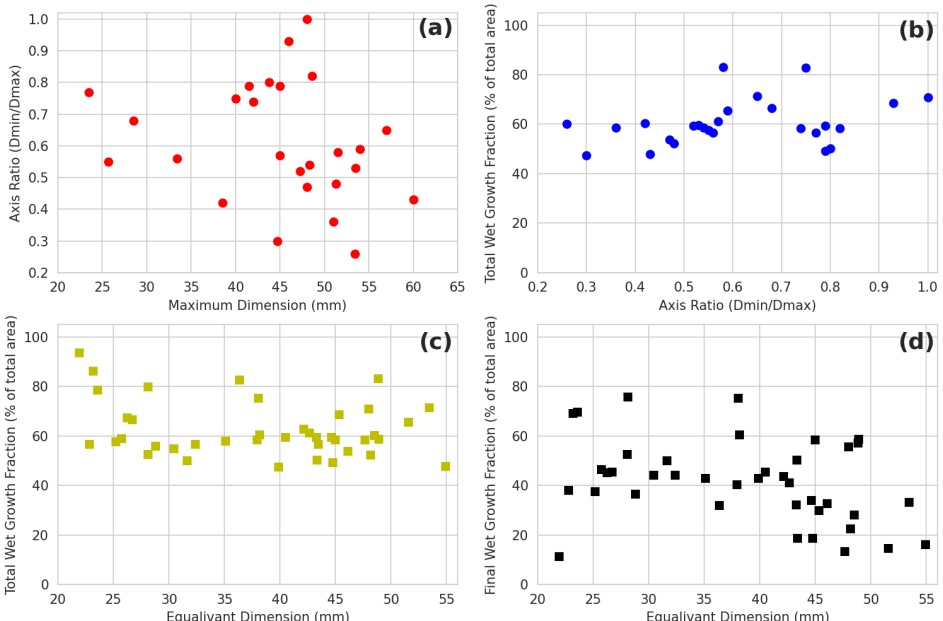

**Figure 6.** Hailstone properties derived from the LAT and manual measurements. Layer analysis technique derived properties include the wet growth across the hailstone cross section as a fraction of the total area and the final wet growth layer as a fraction of the total area. Equivalent dimension was calculated from an oblate spheroid model using the maximum and intermediate dimensions. Note that the minimum dimension used to calculated axis ratio was not measured for all hailstones and therefore panels (a) and (b) have a reduced sample size of 26 compared the complete collection of 40 hailstones used in panels (c) and (d).

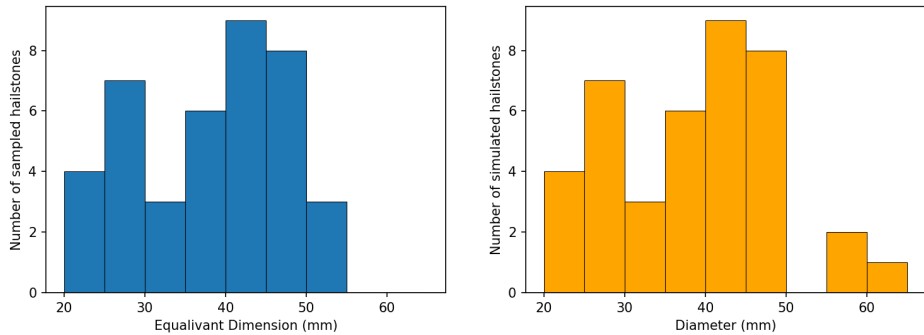

**Figure 7.** Equivalent dimension (oblate spheroid approximation using maximum and intermediate dimensions) distribution for hailstones observed for the Melbourne hailstorm (left) and diameter distribution of those simulated by the hail growth and trajectory model (right).





**Figure 8.** Circular hailstone cross sections generated from the 'umax31' storm of Kumjian and Lombardo (2020). White-shaded regions represents opaque dry-growth layers and gray shaded regions indicate translucent wet-growth layers. Grid lines shown at a 1 cm spacing starting from 0.5 cm from the embryo centroid.





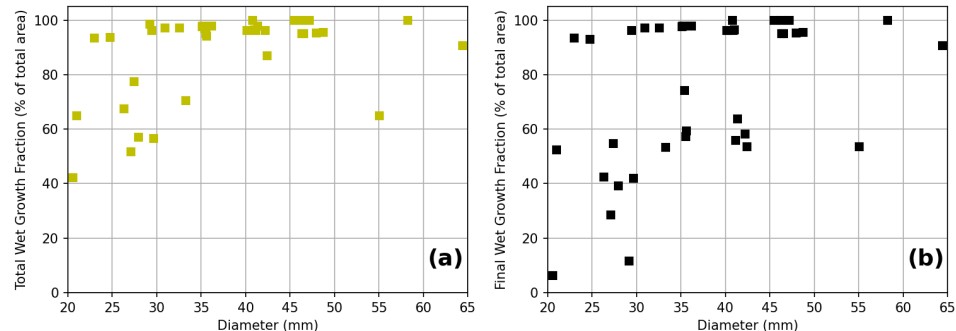

**Figure 9.** Hailstone properties derived from the 'umax31' storm of Kumjian and Lombardo (2020) as a function of diameter. These properties include the wet growth across the hailstone cross section as a fraction of the total area and the final wet growth layer as a fraction of the total area.