# Peer review of "Automating the Analysis of Hailstone Layers"

_EGUsphere, 2022_

## Referee Comment (RC1)

Review of "Automating the Analysis of Hailstone Layers", by Joshua Soderholm and Matthew Kumjian, egusphere-2022-675.

This article present a methodology for scanning a sectioned hailstone to their layered structure. The layered structures inside hailstones provide a direct indication of their shape and properties at various stages during their growth. As temperatures approach $0^0C$, the density of the accreted ice (rime) increases (Macklin, 1962; Pflaum and Pruppacher, 1979). When in wet growth, the component frozen the structure becomes dark, and at the colder temperatures, the structure is lighter. These structures are identified using the scanned images. A total of 40 images were scanned in this study.

The "LAT" observations were compared with model runs which were used to simulate growth layers in a sample of hailstones for comparison with the observed data. As the authors note, the sample of simulated hailstones was selected to approximately match the number and sizes of observed hailstones from the Melbourne case.

The article is well written, and the figures are exceptionally presented. The figures look beautiful and portray the discussion of them. The hailstone scanning procedure is clearly illustrated in Figure 2.

I suggest having a schematic (it would be your Figure 1) from one of your sectioned hailstones (such as below) from your Figure 4, identifying the different regions that can be identified. The discussion you have now is good, but a figure would help.

[Figure]

Section 2.2. Very good discussion. About how long does it take to scan one image?

Regarding the model runs, are the cloud base temperatures, and the subcloud temperature and relative humidity distributions, similar between the model and the 19 January 2020 Melbourne

hailstorm? How are the hailstone densities calculated? Also, an interesting question. Using the model, can you get the density of the water accreted on the hailstones as a function of their diameter? Could you possibly show a figure that would show the radial distribution of the density and particle temperatures for some of your stones?

As noted, there is too much wet growth for the modeled hailstones relative to the observations. The most plausible explanation is that the thermal transfer coefficient (ventilation coefficient) is too low. Perhaps a sensitivity study in the model, increasing the size-dependent ventilation coefficient. Is it possible that if there is wet growth, a "layer" of the dry growth underneath it may become wet, expanding the wet growth thickness?

One other thought. Might it be possible to use the scanned imagery to determine to a first approximation the accreted density? (Search for density measurements using a camera-I found some in Google Chrome). That would be extremely valuable as it would facilitate the estimation of the temperature at which the accretion occurred through use of the Pflaum and Pruppacher (1979) relationship. These temperatures could be compared to those derived from an isotopic analysis of some of the hailstones.

Andy Heymsfield
NCAR
September 23, 2022

---

## Referee Comment (RC2)

**Title**: Automating the analysis of hailstone layers

**Authors**: J. S. Soderholm & M. R. Kumjian

**DOI**: egusphere-2022-675

**General comments:**
This article details a novel methodology for quantitatively and objectively analyzing hailstone growth layers to make inferences about their growth histories and trajectories and which can serve as a way to generate databases which hailstone growth model output can be compared and validated against. The paper is succinct and well-written and will be a valuable contribution to the field, helping to very-much address the large existing gap in hailstone observations and hailstone model validation.  I have no concerns with the manuscript beyond the minor comments below and am glad to see such advancements being made.

**Specific comments:**
L59**:** I am doubtful it would have affected the measurements appreciably, but can the authors include how long the time gap was between when the hailstones were collected and when they were ultimately sliced and photographed? I'm wondering mostly about the potential for (likely minor) sublimational losses while in the freezer, or how that may have affected at least the very outer layer of each hailstone.

L101: Regarding "80" and "25", is this on the 0-255 scale or normalized to a 0-100 scale?

L102: Was the efficacy of the algorithm strongly affected by these find_peaks parameter choices (and, e.g., the 30%-of-peak threshold for consolidating layers) or is it relatively immune to the specific values chosen? The same question goes for the parameters applied to the consolidated smoothed radial on lines 115-117.

L135: Is my understanding correct that the "total wet growth fraction" is the % of cross-sectional area that is due to wet growth, and the "final wet growth layer fraction" the % of the cross-sectional area that is present in the outer-most wet growth layer? (Even if the outer layer is due to dry growth? Or in that case should it be 0%?) This was for some reason a bit confusing to me at first but was made clearer by the caption of Figure 6. Perhaps a brief explainer in-text of what is meant by each term may be helpful to readers.

L139: Is it known how well the oblate spheroid model fits (vs. an ellipsoidal model) for the 26 hailstones that were measured in 3D?

L149: I certainly understand and appreciate the uncertainty in so many of the parameters governing hailstone melting. Nevertheless, is it possible to add a brief sentence about how much melting might be expected under typical conditions for hailstones of different sizes? I'm thinking just an order-of-magnitude-style estimate. E.g., simulations in Ryzhkov et al. (2013a) show that for a 35-mm hailstone over 4-km only about 5 mm of ice core diameter is lost. This

might help orient readers to how severe these impacts from melting might be expected to be regarding the true nature of the outermost layer of these stones.

**Technical corrections:**
L75: Missing closing parenthesis

L87: Move parenthesis to around year

L119: "if" should be "of"

L130: "a" should be "the"

---

## Author Comment (AC1)

Review of "Automating the Analysis of Hailstone Layers", by Joshua Soderholm and Matthew Kumjian, egusphere-2022-675.

**Grapuel Embryo**

This article present a methodology for scanning a sectioned hailstone to their layered structure. The layered structures inside hailstones provide a direct indication of their shape and properties at various stages during their growth. As temperatures approach 0 0 C, the density of the accreted ice (rime) increases (Macklin, 1962; Pflaum and Pruppacher, 1979). When in wet growth, the component frozen the structure becomes dark, and at the colder temperatures, the structure is lighter. These structures are identified using the scanned images. A total of 40 images were scanned in this study. The "LAT" observations were compared with model runs which were used to simulate growth layers in a sample of hailstones for comparison with the observed data. As the authors note, the sample of simulated hailstones was selected to approximately match the number and sizes of observed hailstones from the Melbourne case. The article is well written, and the figures are exceptionally presented. The figures look beautiful and portray the discussion of them. The hailstone scanning procedure is clearly illustrated in Figure 2.

Andy, thank you for taking the time to review this paper and providing us with lots of useful feedback and ideas. Please see below our replies to individual questions in blue coloured text.

• I suggest having a schematic (it would be your Figure 1) from one of your sectioned hailstones (such as below) from your Figure 4, identifying the different regions that can be identified. The discussion you have now is good, but a figure would help.

Thank you for this great idea. A new figure (figure 1) has been added to the manuscript and referenced as part of the introduction to different ice types (line 16), comments on structures within the Melbourne hailstones (line 136) and the discussion of the final wet growth layer in line 161.

• Section 2.2. Very good discussion. About how long does it take to scan one image?

The computation requirements have been added in lines 126-127. It only takes 1-2 seconds to apply the LAT to each image on a low-end desktop computer.

• Regarding the model runs, are the cloud base temperatures, and the subcloud temperature and relative humidity distributions, similar between the model and the 19 January 2020 Melbourne hailstorm?

Thermodynamic indices (LCL, LFC and EL) for three standard parcels (SB, MU and ML) are documented in table 1 and 2 for the umax31 and observed environments, respectively. It's immediately obvious that the umax31 simulation profile has a higher relatively humidity, regardless if the parcel is taken from the SB, MU or ML. Given the SB parcel was also the MU parcel for the case study, and photographs of the storm structure indicate near surface inflow (not shown here), the SB is taken to be the most representative profile. For the SB parcel, the observed LCL and LFC were 510 m and 213 m higher than the umax31 profile, respectively. In terms of the profile temperature at the SB LCL, the observed profile was 2.8 K cooler for the simulation, suggesting comparable cloud base temperatures despite the fact the simulation was not intended to model the Melbourne event.

The manuscript has been updated to indicate that the umax31 sounding indeeds contains more moisture than the observed soundings for the case study in Melbourne (Lines 186-188).

|                                | SB    | MU    | ML (lowest 100 mb) |
|--------------------------------|-------|-------|--------------------|
| LCL (m AGL)                    | 1000  | 1278  | 1057               |
| Profile temperature at LCL (K) | 291.3 | 289.3 | 290.9              |
| LFC (m AGL)                    | 1609  | 1125  | 1563               |

 Table 1: Thermodynamic parameters from the umax31 simulation

Table 2: Thermodynamic parameters from the 1359 LST (two hours prior to hailstorm passage) 19/01/2020 Melbourne Airport sounding

|                                | SB    | MU    | ML (lowest 100 mb) |
|--------------------------------|-------|-------|--------------------|
| LCL (m AGL)                    | 1510  | 1510  | 1591               |
| Profile temperature at LCL (K) | 288.5 | 288.5 | 287.7              |
| LFC (m AGL)                    | 1822  | 1822  | 2080               |

**• How are the hailstone densities calculated?**

The hailstone density is computed following the description in Kumjian & Lombardo (2020, J. Atmos. Sci.), summarized briefly here. For dry growth, accreted rime density is calculated following an adaptation of the Heymsfield and Pflaum (1985) parameterization, with the droplet impact speed reduction (owing to averaging over all impact angles) set to 0.65 (see Rasmussen and Heymsfield 1985). Additionally, we set the minimum allowable density to 500 kg m-3. This latter threshold differs from other implementations, e.g., HAILCAST, and is guided by reasonable ranges of densities reported in Heymsfield et al. (2018). For more details, see Kumjian & Lombardo (2020), pp. 2768-2769.

For wet growth, we follow an adaptation of the parameterization suggested in Rasmussen and Heymsfield (1987) for spongy ice, which is a function of the frozen fraction. Densification through soaking of accumulated liquid water is also treated, following Ziegler et al. (1983) and Rasmussen and Heymsfield (1987). For more details, see Kumjian & Lombardo (2020), pp. 2770-2771.

The density of the mass added in a given timestep is used to add the new layer thickness, and the total average density of the hailstone is updated to re-compute the fallspeed. For more details, see Kumjian & Lombardo (2020), p. 2772, and eqn 24 in that paper.

**• Also, an interesting question. Using the model, can you get the density of the water accreted on the hailstones as a function of their diameter?**

Interesting question, indeed! It was not built into the original KL2020 code, but in principle this information is obtainable. We added some code to track the instantaneous density of ice mass added in each timestep. Keep in mind these do not account for later soaking, which acts to densify the hailstones (and does so quite effectively in our model).

**• Could you possibly show a figure that would show the radial distribution of the density and particle temperatures for some of your stones?**

We can show the density of mass added as a function of particle radius, as well as the particle surface temperature at various times. However, keep in mind this is the instantaneous value, which does not account for soaking and densification. We also show the overall average density as a function of time for the hailstones, but this doesn't reveal the radial distribution.

Given that these are more germane to the model, and not the main focus of the paper, we opted to not show these in the manuscript. We agree, though, that future studies should attempt to obtain estimates of the radial profile of density through real hailstones, and then directly compare this to the model results. (It would require some more coding to get the final radial distribution of density in the hailstones; that is not available at the moment.)

Fig. R1: Hailstone instantaneous added density as a function of size for a sample of the hailstones simulated in the paper. Vertical lines dropping to zero are an artifact of the end of the trajectory (real hailstone densities are limited to 500 kg m-3, as explained above).

Fig. R2: Overall average hailstone density as a function of time for the same sample shown in Fig R1. The colors are consistent.

• As noted, there is too much wet growth for the modeled hailstones relative to the observations. The most plausible explanation is that the thermal transfer coefficient

(ventilation coefficient) is too low. Perhaps a sensitivity study in the model, increasing the size-dependent ventilation coefficient.

This specific sensitivity study wasn't performed in the overview paper of the hail growth and trajectory model (Kumjian and Lombardo 2020). It would certainly be very useful to better understand the dependence of growth regimes on size varying ventilation. I have passed this suggestion onto Matthew Kumjian for his future work.

• Is it possible that if there is wet growth, a "layer" of the dry growth underneath it may become wet, expanding the wet growth thickness?

Thank you for raising this important issue. Prodi et al. 1986 investigated this "two stage growth", whereby initial dry growth is melted and soaked by a wet regime. This type of soaking produces unique features, including large radial bubbles, which are apparent in some wet growth of the Melbourne hailstone collection, confirming that dry growth has very likely been denisified by soaking/melting. A new sentence has been added in lines 138-139 to link the presence of large bubbles to soaking of dry growth.

The hail growth and trajectory model includes considerations for soaking in a bulk sense, whereby unfrozen liquid is soaked until the density of the entire hailstone is equal to solid ice (Kumjian and Lombardo 2020). Given the findings of Prodi et al. 1986, this approach is likely to overestimate the magnitude of soaking, and thereby increase the portion of wet growth beyond that found in natural hailstones. An additional comment has been added to the discussion of model results to highlight the potential role of soaking in excessive wet growth (lines 191-193).

Thank you for this invaluable comment, Andy. It's made us review the role of soaking for the cross sections analysis and in Matt's hail growth and trajectory model.

• One other thought. Might it be possible to use the scanned imagery to determine to a first approximation the accreted density? (Search for density measurements using a camera-I found some in Google Chrome). That would be extremely valuable as it would facilitate the estimation of the temperature at which the accretion occurred through use of the Pflaum and Pruppacher (1979) relationship. These temperatures could be compared to those derived from an isotopic analysis of some of the hailstones.

Yes, this idea has crossed my mind to pursue the application of photomicroscopy to determine bubble concentration by Carras and Macklin 1975. I think there are two ways to approach this – take the local mean of lightness field with a known cross section thickness OR capture a small portion of the cross section at a sufficiently high resolution to resolve individual bubbles (maybe 1x1cm at 50MP?). Both techniques would require a reference measurement of the true density though, so it might be best to start with artificial ice layers. I have added this idea to the conclusion in lines 212-214.

Andy Heymsfield NCAR September 23,

---

## Author Comment (AC2)

Title: Automating the analysis of hailstone layers

Authors: J. S. Soderholm & M. R. Kumjian

DOI: egusphere-2022-675

**General comments:**

This article details a novel methodology for quantitatively and objectively analyzing hailstone growth layers to make inferences about their growth histories and trajectories and which can serve as a way to generate databases which hailstone growth model output can be compared and validated against. The paper is succinct and well-written and will be a valuable contribution to the field, helping to very-much address the large existing gap in hailstone observations and hailstone model validation. I have no concerns with the manuscript beyond the minor comments below and am glad to see such advancements being made.

Jacob, thank you for taking the time to review this paper and providing us with lots of useful feedback and ideas. Please see below our replies to individual comments in blue coloured text.

**Specific comments:**

L59: I am doubtful it would have affected the measurements appreciably, but can the authors include how long the time gap was between when the hailstones were collected and when they were ultimately sliced and photographed? I'm wondering mostly about the potential for (likely minor) sublimational losses while in the freezer, or how that may have affected at least the very outer layer of each hailstone.

Laboratory analysis wasn't performed immediately after collection, rather is was closer to three months afterwards. Hailstones were individually sealed inside plastic bags to limit sublimation, but some losses would have still occurred. The information has been added to the manuscript in lines 59-61.

L101: Regarding "80" and "25", is this on the 0-255 scale or normalized to a 0-100 scale?

This sentence has been expanded to clearly state that these values are on the 0-255 scale. Please see lines 102-105.

L102: Was the efficacy of the algorithm strongly affected by these find\_peaks parameter choices (and, e.g., the 30%-of-peak threshold for consolidating layers) or is it relatively immune to the specific values chosen? The same question goes for the parameters applied to the consolidated smoothed radial on lines 115-117.

The find\_peaks parameters are closely tied to physical properties of imagery, and thus are sensitive to changes in the values. For example, to detect dry growth layers, the local maxima threshold was manually selected such that the (stretched) lightness value is the transition between wet and dry growth, the prominence represents the minimum observed difference in lightness between wet and dry growth and the separation was selected as the minimum separation of dry growth layers that could be robustly resolved in the imagery. The methodology section has been expanded in lines 102-109 to document the physical dependence of the parameters and whether they need to be retuned for new datasets.

The second application of the find\_peaks function is less sensitive to the choice of parameters, but would still require re-optimisation if the image resolution or bin width used in the analysis technique changed. This additional information has been added to the lines 124-125.

L135: Is my understanding correct that the "total wet growth fraction" is the % of cross-sectional area that is due to wet growth, and the "final wet growth layer fraction" the % of the cross-sectional area that is present in the outer-most wet growth layer? (Even if the outer layer is due to dry growth? Or in that case should it be 0%?) This was for some reason a bit confusing to me at first but was made clearer by the caption of Figure 6. Perhaps a brief explainer in-text of what is meant by each term may be helpful to readers.

Thank you for this suggestion. An explanation of these two metrics has been added to the text in lines 145-147.

L139: Is it known how well the oblate spheroid model fits (vs. an ellipsoidal model) for the 26 hailstones that were measured in 3D?

The mean difference between the equivalent dimension calculated from the spheroid and ellipsoidal models was -7.3 mm (standard deviation of 4.5 mm), indicating the lower-order oblate spheroid model has a bias towards larger equivalent dimension for this dataset (as you'd expect from the poorer fit). While this figure is somewhat large (~20% of the mean Deq), the consistent sign of the bias suggests the results can still be interpreted with some confidence. I've documented the mean bias in line 151 for the reader.

L149: I certainly understand and appreciate the uncertainty in so many of the parameters governing hailstone melting. Nevertheless, is it possible to add a brief sentence about how much melting might be expected under typical conditions for hailstones of different sizes? I'm thinking just an order-of-magnitude-style estimate. E.g., simulations in Ryzhkov et al. (2013a) show that for a 35-mm hailstone over 4-km only about 5 mm of ice core diameter is lost. This might help orient readers to how severe these impacts from melting might be expected to be regarding the true nature of the outermost layer of these stones.

Thank you for this suggestion. I can see the value in this, especially given the smaller hailstones in the collection. We've added this new information into lines 163-165.

**Technical corrections:**

L75: Missing closing parenthesis

L87: Move parenthesis to around year

L119: "if" should be "of"

L130: "a" should be "the"

Thank you for picking up these corrections. They have all been amended in the text.